# Gradual modality dropout for segmenting ischemic stroke lesions in an unseen center with missing modalities

**Sofia Vargas-Ibarra**[1]                                                   SOFIA.VARGASIBARRA@UNIV-EVRY.FR
**Vincent Vigneron**[1]                                                     VINCENT.VIGNERON@UNIV-EVRY.FR
**Hichem Maaref**[1]                                                        HICHEM.MAAREF@UNIV-EVRY.FR
[1]*Université Évry Paris-Saclay, IBISC, Évry, France.*

**Sonia Garcia-Salicetti**[2]                                    SONIA.GARCIA@TELECOM-SUDPARIS.EU
[2]*SAMOVAR, Télécom SudParis, Institut Polytechnique de Paris, Palaiseau, France.*

**Andreia Faria**[3]                                                               AFARIA1@JHMI.EDU
[3]*Johns Hopkins University, Baltimore, USA.*

**Editors:** Accepted for publication at MIDL 2025

## Abstract

In clinical practice, imaging modalities may not always be available for every patient due to scheduling, cost, or patient-specific constraints. Additionally, multi-center imaging studies often face inconsistencies in protocols, machine settings, and artifacts, compromising data quality. We propose a 3D U-Net model for ischemic lesion segmentation using a novel training technique, gradual modality dropout, which progressively deactivates imaging modalities during training. This approach ensures robust performances when all modalities are present and improves segmentation accuracy in scenarios where one or more modalities are missing in unfamiliar contexts. The model demonstrates adaptability and reliability when trained on MRI scans of stroke patients across different phases (hyper-acute, sub-acute, acute, and post-treatment) and various hospital settings. Code available here: https://github.com/sofiavarib/Gradual-modality-dropout

**Keywords:** Stroke, segmentation, MRI modalities, lesion, missing modalities

## 1. Introduction

Stroke remains a leading cause of death globally, with millions of fatalities each year. It results from arterial blockages that reduce blood flow, causing irreversible brain damage. Accurate identification of the affected brain region (lesion) is vital for treatment decisions. While lesion segmentation is critical in improving survival and recovery, manual analysis is time-consuming and limits timely intervention.

To address this, various deep learning models have been developed to automate and accelerate lesion detection. These include 3D CNN-based models like 3D U-Net (Ronneberger et al., 2015; Omarov et al., 2022; Ashtari et al., 2023), hybrid 2D/3D architectures such as D-Unet (Zhou et al., 2021) and DFENet (Basak et al., 2021), patch-based sampling (Xue et al., 2020; Alquhayz et al., 2022), and attention-based methods like AABTS-Net (Tian et al., 2023) and PerfUnet (de Vries et al., 2023). However, many rely on the presence of specific imaging modalities. To overcome this, approaches like the Unified Representation Network (Lau et al., 2019) enable flexible input handling, while GANs have been used to generate missing modalities (Sharma and Hamarneh, 2019). Modality dropout strategies have also proven effective, with MultiUnet (Xu et al., 2024) and ModDrop+ (Liu et al., 2022) offering strong performance, the latter introducing a dynamic filter-scaling head.

## 2. Datasets

This study uses two multicentric and multiequipement datasets: JHU (Liu et al., 2023), which consists of 1888 patients with acute ischemic strokes, and ISLES (Hernandez Petzsche et al., 2022), which has 150 patients. CHSF and MATAR datasets (Marnat et al., 2023) consist only of hyper-acute strokes from a single hospital and machine. CHSF has 65 patients with proximal occlusion and MATAR 125 patients with distal ones. All of them have Apparent diffusion coefficient (ADC), magnetic field strength (B0), and diffusion-weighted imaging (DWI), except for ISLE,S where B0 is missing.

## 3. Method

Inspired by the dropout technique (Srivastava et al., 2014), modality dropout (Lau et al., 2019) extends this concept to input modalities (Lau et al., 2019): instead of deactivating individual neurons, it zeroes out entire input images with a Bernoulli probability $p$, effectively simulating missing modalities and improving model robustness to incomplete data. We propose a gradual transition mechanism to replace the abrupt disruptions caused by completely dropping a modality, and evaluate its performance using the nnU-Net benchmark (Isensee et al., 2018). For a segmentation model $F$ that processes input modalities $\mathbf{x}_j, j \in \{1, \ldots, n\}$ and $n$ the total number of modalities, the final output $\mathbf{y}$ is computed as:

$$\mathbf{y} = F(\widetilde{\mathbf{r}} \odot \mathbf{x}) \quad \text{with} \quad \widetilde{\mathbf{r}}_j = \begin{cases} 1, & \text{if } \mathbf{r}_j = 1, \text{with probability } p \\ g_j(t), & \text{otherwise, with probability } 1 - p \end{cases} \tag{1}$$

with $\mathbf{r}_j \sim \text{Bernouilli}(p)$ and '$\odot$' the Hadamard product. When $p = 1$, no dropout is applied, and all modalities are used, whereas when $p = 0$, the corresponding modality is gradually dropout, controlled by the gradual function $g(t)$. This decreasing function reaches zero at some time $t$ and we propose the following $g(t)$:

$$g_j(t) = \begin{cases} 0.75 + \epsilon, & \text{if } t < 0.25T \\ 0.5 + \epsilon, & \text{if } t < 0.5T \\ 0.25 + \epsilon, & \text{if } t < 0.75T \\ 0 + \epsilon, & \text{otherwise} \end{cases} \tag{2}$$

where $T$ represents the total number of epochs, and a noise $\epsilon$ is added from a normal distribution with a mean of 0 and a standard deviation of 0.01 to the function $g_j(t)$. Serving both as data augmentation, where $\widetilde{\mathbf{r}} \odot \mathbf{x}$ becomes an enhanced version of $\mathbf{x}$, and as regularization, it allows the model to adapt to missing modality data gradually.

## 4. Results

In clinical practice, a dataset from a new center can create a missing modality scenario. To mimic this case, all datasets are used during training except ISLES, where B0 is absent. Using ADC, DWI, and B0 as inputs for nnUnet, we apply modality dropout on B0 with varying probabilities and evaluate performance with all modalities available and under missing-data conditions. The results are shown in Table 1, where our proposed method

**Table 1:** The evaluation considers missing B0 on the unseen ISLES dataset, using modality dropout with probability $p$ in gradual and non-gradual settings. All training configurations use ADC, DWI, and B0 as inputs. The upper bound (*) includes ISLES in training, while the lower bound (†) excludes it, both without any method. The best Dice scores for unseen datasets and average results under missing-modality conditions are highlighted and the best without absent data are underlined. Evaluation symbols: ● for all modalities present, and ○ when B0 is absent.

| Exp. num | Mod. drop Type | 1-$p$ | B0 | Unseen ISLES | Test set ($T_0$) (Dice) CHSF | MATAR | ISLES (unseen) | JHU | Average |
|---|---|---|---|---|---|---|---|---|---|
| Exp1 | - | - | ● | ✓ | 0.790± 0.029 | 0.768±0.061 | - | 0.731±0.012 | 0.763±0.059 |
| Exp1 | - | - | ○ | ✓ | 0.787±0.027 | 0.767±0.065 | $0.622^{\dagger}$±0.040 | 0.723±0.014 | 0.725±0.055 |
| Exp2 | MultiUnet | 0.2 | ● | ✓ | 0.787±0.027 | 0.762±0.056 | - | 0.750±0.012 | 0.767±0.061 |
| Exp2 | MultiUnet | 0.2 | ○ | ✓ | 0.788±0.028 | 0.765±0.056 | 0.672±0.039 | 0.748±0.013 | 0.743±0.051 |
| Exp3 | Gradual | 0.2 | ● | ✓ | 0.789±0.028 | 0.757±0.061 | - | 0.744±0.013 | 0.763±0.061 |
| Exp3 | Gradual | 0.2 | ○ | ✓ | 0.789±0.028 | 0.758±0.061 | 0.650±0.040 | 0.743±0.013 | 0.735±0.052 |
| Exp4 | ModDrop+ | 0.2 | ● | ✓ | 0.787±0.028 | 0.733±0.066 | - | 0.740±0.014 | 0.753±0.066 |
| Exp4 | ModDrop+ | 0.2 | ○ | ✓ | 0.787±0.029 | 0.732±0.065 | 0.617±0.041 | 0.740±0.014 | 0.719±0.056 |
| Exp5 | MultiUnet | 0.5 | ● | ✓ | 0.789±0.028 | 0.775±0.056 | - | 0.748±0.013 | 0.771±0.058 |
| Exp5 | MultiUnet | 0.5 | ○ | ✓ | 0.789±0.028 | **0.777±0.055** | 0.637±0.040 | 0.747±0.013 | 0.738±0.051 |
| Exp6 | Gradual | 0.5 | ● | ✓ | 0.786±0.029 | 0.766±0.055 | - | 0.747±0.013 | 0.766±0.058 |
| Exp6 | Gradual | 0.5 | ○ | ✓ | 0.786±0.029 | 0.763±0.056 | 0.655±0.039 | 0.748±0.012 | 0.738±0.050 |
| Exp7 | ModDrop+ | 0.5 | ● | ✓ | 0.779±0.029 | 0.758±0.070 | - | 0.744±0.013 | 0.760±0.066 |
| Exp7 | ModDrop+ | 0.5 | ○ | ✓ | 0.778±0.029 | 0.761±0.070 | 0.630±0.040 | 0.743±0.013 | 0.728±0.056 |
| Exp8 | MultiUnet | 0.8 | ● | ✓ | 0.790±0.026 | 0.743±0.068 | - | 0.751±0.013 | 0.761±0.065 |
| Exp8 | MultiUnet | 0.8 | ○ | ✓ | 0.790±0.027 | 0.741±0.068 | 0.682±0.032 | 0.751±0.013 | 0.741±0.050 |
| Exp9 | Gradual | 0.8 | ● | ✓ | 0.810±0.025 | 0.750±0.055 | - | 0.743±0.013 | 0.768±0.057 |
| Exp9 | Gradual | 0.8 | ○ | ✓ | **0.810±0.026** | 0.752±0.056 | **0.701±0.038** | 0.743±0.013 | **0.749±0.048** |
| Exp10 | ModDrop+ | 0.8 | ● | ✓ | 0.788±0.026 | 0.763±0.060 | - | 0.753±0.012 | 0.768±0.058 |
| Exp10 | ModDrop+ | 0.8 | ○ | ✓ | 0.788±0.027 | 0.761±0.060 | 0.635±0.038 | **0.753±0.012** | 0.734±0.051 |
| Exp11 | - | - | ● | | 0.773±0.031 | 0.753±0.056 | - | 0.756±0.012 | 0.761±0.057 |
| Exp11 | - | - | ○ | | 0.775±0.025 | 0.680±0.058 | $0.782^{*}$±0.029 | 0.756±0.012 | 0.747±0.041 |

(Gradual) is compared with MultiUnet, where the modality dropout is done without a gradual function, zeroing abruptly the modalities and with ModDrop+, where a dynamic filter is included. Some visual predictions are shown in Appendix A.

Except for ModDrop+ with $(1-p) = 0.2$, all methods allow to have an improvement on the unseen dataset ISLES with respect to the lower bound (which corresponds to training without ISLES and no dropout (Exp1)). The best result is obtained with our proposed method, arriving at a Dice of 0.701 on ISLES (Exp 9), but still staying far from the upper bound, which includes ISLES during training with a black image instead of B0 input (Exp11, Dice = 0.782). Seeing the average Dice scores under missing modalities conditions, the best average score (Exp9, Dice 0.749) gets close to the upper bound (Exp11, Dice 0.747), without a performance reduction under the non-missing scenario, showing the method's robustness.

## 5. Conclusion

The proposed method includes modality dropout in a smoother way, where a gradual function reduces the input contrast over the training. Thanks to the regularizing and augmentation aspect of the method and without reducing the performance in the seen centers either under a missing or not missing scenario, it outperforms the other Dice score' methods on an unseen center, where a modality is absent. Gradual modality dropout is model-agnostic, being adaptable to any architecture under any training schedule.

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

## Appendix A. Out-domain evaluation with missing modalities - visual examples

In Fig. 1, the predictions of several models and the groundtruth are represented over the DWI image for ISLES dataset. All models have DWI, ADC, and missing B0 as input (black image). For the first column, the lower bound model is used, when ISLES is not seen during training, the second one is the best model obtained by adding the proposed method $(1 - p = 0.8)$, and the third column is when the model uses ISLES during training. Finally, in the last column, the groundtruth of the model is shown. In the first line, when the dataset is unseen and no method to manage the missing modality is used, some parts of the lesion are not detected. For the second case, false positives are obtained if the method is not applied. Finally, in the last case, we can see how seeing or not seeing ISLES, there are parts of the lesion that are still not detected.

| Unseen ISLES | Unseen ISLES + grad. mod. drop. | Seen ISLES | Groundtruth |
| --- | --- | --- | --- |

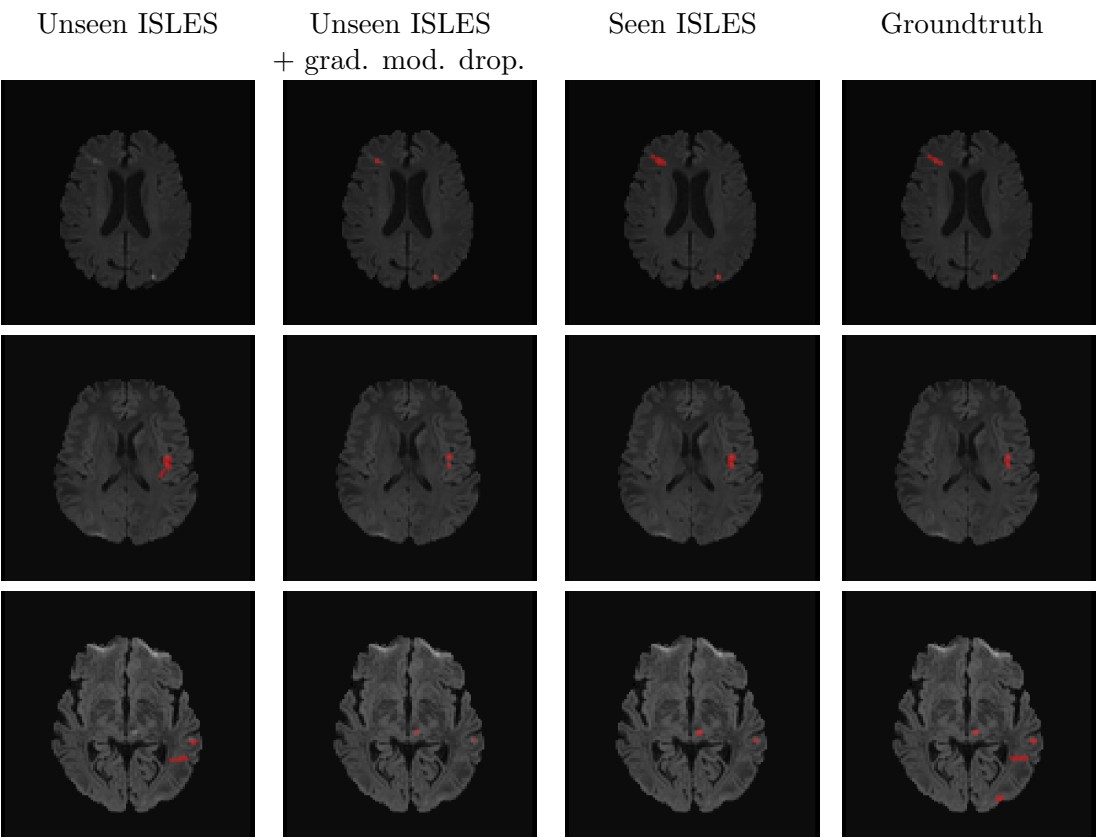

**Figure 1:** Lesion segmentation from ISLES dataset using ADC, DWI and missing B0 (black image)

