# OpenReview forum: "Gradual modality dropout for segmenting ischemic stroke lesions in an unseen center with missing modalities"
_MIDL.io/2025/Short_Papers — MIDL 2025 - Short Papers_

### Official Review · Reviewer_n4az · 2025-04-28

**Rating:** 4
**Confidence:** 4

**Summary:**

The paper proposes a technique to make multi-modal stroke segmentation more robust to missing modalities. It randomly replaces modalities by lower-intensity versions (plus noise). The intensity values decrease more as training progresses. This is meant as a softer version of modality dropout.

**Strengths:**

- Code is provided
- 4 different datasets are used, including several MRI models, resulting in a large cohort to test. This is a very strong point of the paper.
- experiments with multiple values for modality dropout probability are performed
- several baselines are used (MultiUnet, ModDrop+), with convincing results on unseen dataset (ISLES) and slightly better results on average than the baseline (although the difference is probably not significant)

**Weaknesses:**

- The approach proposed changes the dynamic range of data, modality-wise. The interplay between the proposed approach and preprocessing (overall intensity normalisation) as well as batch normalization, and its putative impact on training dynamics (perhaps through changing magnitude of gradients?), should be explored.

---

### Decision · Program_Chairs · 2025-05-01

Accept